# Unpacking the Drivers of Dissatisfaction and Satisfaction in a Fitness Mobile Application

**DOI:** 10.3390/bs13090782

**Published:** 2023-09-21

**Authors:** Minseong Kim, Sae-Mi Lee

**Affiliations:** 1Department of Management & Marketing, College of Business, Louisiana State University Shreveport, Shreveport, LA 71115, USA; minseong.kim@lsus.edu; 2School of Global Business, Kyungil University, 50 Gamasil-gil, Hayang-eup, Gyeongbuk, Gyeongsan-si 38428, Republic of Korea

**Keywords:** physical activity, fitness, technology, two-factor model, mobile application

## Abstract

This research investigates the factors influencing user satisfaction and dissatisfaction in fitness mobile applications. It employs Herzberg’s two-factor model through text mining to classify Fitbit mobile app attributes into satisfiers and dissatisfiers. The Fitbit app was chosen due to its prevalence in the United States. The study analyzes 100,000 English reviews from the Fitbit app on the Google Play Store, categorizing attributes. It identifies three dissatisfying categories (functional, compatibility, paid services) and three satisfying categories (gratification, self-monitoring, self-regulation), comprising 25 sub-attributes. This classification offers in-depth insights into what drives user contentment or discontent with fitness apps. The findings contribute to the fitness app domain by applying text-mining and Herzberg’s model. Researchers can build upon this foundation, and practitioners can use it to enhance app experiences. However, this research relies on user reviews, often lacking comprehensive explanations. This limitation may hinder a profound understanding of the underlying psychological aspects in user sentiments. Nonetheless, this study takes strides toward optimizing fitness apps for users and developers.

## 1. Introduction

The present-day sedentary lifestyle, stress, anxiety, and an unhealthy food environment have become the leading health risks across the world [1]. As a result, people are increasingly motivated to improve their quality of life and overall well-being by engaging in physical activity [2,3]. To facilitate this trend, the Centers for Disease Control and Prevention has introduced physical activity (PA) guidelines aimed at supporting people in their physical activity pursuits [4]. Despite these efforts and the growing trend, research indicates that three out of four adults in the United States fail to meet the recommended physical activity guidelines, which specify engaging in at least 150 min of moderate-intensity physical activity (such as brisk walking) or 75 min of vigorous-intensity physical activity (such as running) per week [4]. Nevertheless, the problem of physical inactivity has become increasingly prevalent in spite of the fact that many people are concerned about the condition of their health. In the United States, sedentary lifestyles, technology-oriented convenience environments, and unhealthy food environments have become leading health risks along with physical inactivity [1,4]. Physical inactivity has been considered to be a serious social issue since it results in severe health problems, including obesity, cardiovascular diseases, diabetes, and mental health disorders. Accordingly, individuals are increasingly motivated to improve their quality of life by engaging in physical activity [2,5].

This research explores the potential of technological innovations, such as smartphones and mobile applications (apps), to support physical activity among individuals. These technologies are highly popular and integrated into daily life [6,7]. Traditional approaches to physical activity have several limitations in terms of available resources, relying primarily on personal trainers or access to specific equipment/facilities that require experts to operate. In addition, traditional approaches to physical activity have low levels of capacities to consider individuals’ diverse abilities, interests, and lifestyles when designing exercise programs [7]. Furthermore, compared to fitness mobile apps, traditional approaches to physical activity could not provide real-time feedback and monitoring as well as social support and accountability (or social connections and supportive online communities) [3]. By addressing these limitations of traditional approaches to physical activity, the development of fitness apps and wearable devices has grown alongside advances in smartphones and mobile apps. These technologies are designed to encourage individuals to exercise and to monitor their physical performance in real time. Fitbit, a well-known fitness app, is an example of an app that offers synchronous tracking of physiological data, such as steps, heart rate, distance, and calorie count, to help individuals optimize their workout routines [8]. Virtual rewards and competition in fitness apps can also have a positive impact on physical activity-related behavior [7,9,10]. According to market research by Grand View Research [11], the global fitness app industry was valued at $2.4 billion in 2018 and is expected to reach $10.9 billion by 2026 due to technological advancements and increasing interest in physical and mental health. 

Despite the widespread use of fitness apps and their contemporary significance, there has been limited research on the features of these apps and users’ engagement with them [5,12,13]. Some studies have conducted systematic reviews of mobile fitness apps with a focus on identifying the motivations for using these apps for physical activity. For instance, Muntaner-Mas et al. [14] reviewed five previous studies and analyzed 88 fitness apps to explore their features and prospects. Hosseinpour and Terlutter [10] conducted a detailed literature review and classified the motivators of mobile phone apps as perceptions of apps and app techniques, such as feedback, goal-setting, competition, and social sharing. Asimakopoulos, Asimakopoulos, and Spillers [15] examined the functional aspects of fitness apps and found that data, gamification, and content facilitated users’ motivation and efficacy. However, little research has focused on the functions and attributes of fitness apps from the users’ perspective, and user satisfaction has been overlooked as a crucial factor that can influence users’ engagement with fitness apps. Motivation alone may not predict the sustained use of fitness apps for physical activity, as users may discontinue using the app if they are not satisfied with it. In the sports medicine field, where sustainable health behaviors and physical and mental well-being are essential, it is imperative to identify the app factors that drive user satisfaction and dissatisfaction from their perspective [16,17].

By applying a text-mining approach based on Herzberg’s two-factor theory [18], the main objective of our research is to exploratorily identify and categorize the digital features and attributes of the Fitbit mobile app that contribute to user satisfaction or dissatisfaction. To achieve this objective, this research employs content analysis of online reviews and ratings collected from the Google store. The results of this study offer critical insights for fitness mobile app developers, helping them optimize users’ satisfaction and minimize dissatisfaction, thereby enhancing the experience of individuals who engage in physical activity through fitness apps. In addition, from a practical standpoint, this study can provide directions for the development of more effective physical activity programs, going beyond traditional approaches and external motivations among fitness mobile app users. By considering the identified satisfiers and dissatisfiers, fitness mobile apps can be tailored to better motivate users and increase their adherence to regular workouts. Also, by understanding the specified satisfiers and dissatisfiers, policymakers and health organizations can promote the integration of mobile app technology-driven solutions into public health strategies. This may be able to reach a larger population, provide personalized support, and encourage users to adopt and maintain a physically active lifestyle in the long-term. 

## 2. Conceptual Background

### 2.1. Herzberg’s Two-Factor Model as a Theoretical Background

Herzberg’s two-factor model emphasizes that satisfaction and dissatisfaction are influenced by the environment [18]. The theory suggests that satisfaction and dissatisfaction are independent and distinct constructs, affected by various motivation and hygiene factors. Motivation factors represent emotional needs that can lead to satisfaction, such as growth, responsibility, achievement, and recognition [19,20]. On the other hand, hygiene factors refer to contextual features presented in the environment that address individuals’ basic needs. An absence of hygiene factors will result in dissatisfaction, but their presence will not increase satisfaction levels. This theory has been widely used in computer and mobile studies to investigate the quality of online platforms like mobile apps, websites, and cloud services [19,20,21,22]. For example, Lee et al. [19] discovered that satisfiers, such as information quality, had a positive effect on mobile data service usage, while dissatisfiers, such as system quality, had a negative effect.

The prevalence of interactive technology has made it an attractive tool for promoting a healthy lifestyle. Among these technologies, mobile devices and fitness apps offer a high degree of customization and independence for users [9,13,23]. Moreover, fitness apps offer personalized training and skill assessment during physical activity [24,25]. Online user reviews have become a crucial source of brand perception that can influence the decisions of both organizations and users [26]. However, little attention has been paid to both user satisfaction and dissatisfaction with fitness mobile apps, despite the increased interest in online reviews across various platforms. For example, Ahn and Park [27] identified motivations for user satisfaction of fitness mobile apps with an online review analysis of user experience. However, their study focused primarily on predicting user satisfaction from pre-specified factors determined by them (i.e., computation of ratio for the factors in all reviews, such as usability, usefulness, affection, and perceived values). In addition, Yin et al. [28] investigated the impact of gamification elements on user satisfaction in fitness and health mobile apps. However, their empirical study analyzed online reviews to identify four dimensions of gamification elements with their intervention during the data analysis stage (i.e., progress path, feedback and reward, social connection, and interface and user experience). Based on these dimensions, their research calculated fitness mobile app users’ satisfaction by neglecting other aspects of the user experience, such as dissatisfaction. Furthermore, Wu et al. [29] pointed out the important role of online reviews in predicting fitness mobile app user satisfaction. Even so, their study measured the construct of online reviews via a survey approach instead of directly analyzing online review data to predict satisfaction and continued usage behavior. To fill the academic gap in this field, using Herzberg’s two-factor model, this study aims to identify the key factors that influence the user experience and that can increase or decrease satisfaction with fitness mobile apps without a researcher’s intervention.

### 2.2. Application of Herzberg’s Two-Factor Model to the Context of Fitness Mobile Apps

Herzberg’s two-factor model, originally developed for the workplace, offers a valuable theoretical framework for the examination of user satisfaction and dissatisfaction within the domain of mobile apps [30,31,32]. This model, when applied in this context, underscores several pertinent aspects. Firstly, Herzberg’s model advocates for a user-centric approach, emphasizing the importance of considering the experiences, preferences, and feedback of fitness app users. Through the analysis of user reviews and ratings, researchers can gain meaningful insights into the specific app attributes that contribute to either user satisfaction or dissatisfaction, aligning closely with the user-focused orientation of Herzberg’s theory [31]. Moreover, the model posits that genuine satisfaction and motivation stem from intrinsic factors, such as a sense of achievement, recognition, and the inherent enjoyment derived from the task itself. In the context of fitness apps, the cultivation of intrinsic motivation is pivotal for sustaining user engagement [30]. A profound understanding of which features and attributes tap into users’ intrinsic motivation can guide the development of more effective fitness apps, enhancing their ability to motivate and engage users effectively [32]. Additionally, Herzberg’s model underscores the distinction between hygiene factors and motivators. While hygiene factors prevent dissatisfaction, they do not inherently inspire motivation. Applied to fitness apps, addressing dissatisfiers like technical glitches or inadequate user support may prevent users from abandoning the app due to frustration. However, fostering true engagement and enduring usage relies on the presence of motivators, which can actively encourage users to adhere to regular exercise routines and attain their fitness objectives [30]. Lastly, Herzberg’s model incorporates both objective and subjective dimensions for evaluating satisfaction and dissatisfaction. In the context of fitness apps, objective metrics encompass aspects like app usage statistics, whereas subjective measures encompass user reviews, ratings, and feedback [31]. This comprehensive approach enables a thorough understanding of user experiences [32]. In conclusion, the application of Herzberg’s two-factor model to the study of fitness apps allows scholars to establish a theoretical framework that not only facilitates the active engagement and motivation of users in their pursuit of improved physical well-being but also prevents user dissatisfaction from a psychological perspective. Through the identification and classification of pertinent factors, researchers can contribute valuable insights to the enhancement of fitness app design and user experiences. 

### 2.3. The Significance of Online Reviews in Consumer Behavior

Our study aims to offer a comprehensive exploration of why Herzberg’s two-factor model serves as an appropriate framework for the investigation of user satisfaction and dissatisfaction within the context of fitness mobile applications. Specifically, a critical element in comprehending user satisfaction and dissatisfaction is the influence exerted by online reviews [33,34]. In today’s digital landscape, where information is effortlessly accessible, online reviews have emerged as influential drivers of consumer behavior [30]. These reviews, authored by individuals who have interacted with diverse products and services, wield substantial sway over prospective consumers. They represent a valuable information source, providing insights into actual user experiences and impacting their decision-making processes [31]. This, in turn, contributes significantly to the establishment of trust and credibility, thereby encouraging increased adoption. Furthermore, online reviews provide a platform for users to openly express their contentment or discontentment [34]. This expression of sentiment not only shapes individual choices but also establishes a feedback mechanism for product developers and service providers. By closely analyzing the content of these reviews, researchers and developers can gain invaluable insights into the aspects of products that resonate with users and those that require enhancement [34]. The wealth of user-generated feedback serves as a valuable repository of actionable data that can guide iterative product development and refinement strategies [31]. Moreover, the influence of online reviews extends beyond individual product decisions and brand reputation [33]; it has evolved into a major catalyst for competition and innovation across various industries, including fitness mobile applications. Companies have grown increasingly attentive to the feedback conveyed in reviews, recognizing that it often highlights areas primed for improvement and innovation. For instance, users might express dissatisfaction with a fitness app’s absence of social integration for virtual workouts. In response to such feedback, developers may prioritize the integration of real-time social features, thereby cultivating a more engaging user experience. 

In summary, online reviews play a pivotal role in bridging the gap between user satisfaction or dissatisfaction and the iterative product development cycles. Through the comprehensive analysis of review content and sentiment, both scholars and industry practitioners can develop a deeper understanding of user preferences and needs. Within the context of our research, the integration of Herzberg’s model in tandem with online reviews offers the promise of delivering a holistic perspective on user satisfaction and dissatisfaction within the domains of computer science and fitness mobile applications. Ultimately, this endeavor contributes to the advancement of technology solutions that are not only more effective but also tailored to meet user-centric requirements.

## 3. Method and Results

### 3.1. Unit of Analysis and Data Collection Procedures

In this research, a Python programming approach was employed to gather a dataset comprising 100,000 English reviews of the Fitbit app. These reviews were voluntarily submitted by users on the Google Play Store. The data collection process followed the subsequent steps: (1) the primary researcher initiated the procedure by installing essential Python libraries designed for web scraping and data manipulation. These libraries encompassed requests, beautifulsoup4, and pandas; (2) to access the Fitbit app’s page on the Google Play Store, the lead researcher utilized Python’s requests library, executing an HTTP GET request directed at the app’s URL; (3) once the HTML content of the Fitbit app page on the Google Play Store was retrieved, the lead researcher utilized beautifulsoup4 for parsing purposes, facilitating the extraction of pertinent information; (4) the lead researcher identified the section housing user reviews by inspecting the HTML structure of the Fitbit app page on the Google Play Store. Through this process, review data, encompassing the review texts, was extracted from the HTML content; and (5) the dataset’s finalization stage involved pagination through the reviews, data cleaning procedures to eliminate duplicates and irrelevant entries, and ultimately storing the dataset as a CSV file to facilitate subsequent analysis.

Researchers and developers can derive technological advantages from utilizing review data sourced from the Google Play Store in contrast to other application marketplaces. This preference is rooted in the subsequent factors [35,36]: (1) timely data availability: Google Play Store reviews are frequently accessible in real time or near real time, facilitating the analysis of user feedback and app performance with the most current data; this immediacy proves invaluable for monitoring app sentiment and promptly addressing emerging issues or trends. (2) An abundance of data: the Google Play Store, boasting an extensive user base and an extensive collection of applications, generates substantial quantities of review data. This abundance enables researchers to work with sizable datasets, enabling more comprehensive analyses and the utilization of advanced data analytics methodologies. (3) Response examination: within the Google Play Store environment, numerous app developers actively engage with users by responding to their reviews; researchers can scrutinize these interactions, monitor developer responses over time, and evaluate their impact on user satisfaction levels and app ratings. (4) Leveraging natural language processing: Google Play Store reviews frequently furnish copious textual data through user comments; researchers can harness natural language processing techniques to scrutinize the sentiment, thematic content, and expressions conveyed in these reviews. This analytical approach provides valuable insights into user sentiments and feedback. The decision to utilize review data in this research was preferred over conducting an online survey to mitigate potential biases arising from user intentions and response consistency [16]. 

### 3.2. Fitbit as a Focus of This Research

The choice of the Fitbit app as the focal point of this study is grounded in its widespread popularity and significant influence on fitness-related technology, coupled with a pronounced interest in wellness and health matters within the United States [9,10]. To elaborate further: first, the Fitbit app boasts an extensive array of health and fitness tracking features that extend well beyond simple workout logging. It adeptly monitors various health metrics such as step counts, heart rate, sleep patterns, and calorie intake. This holistic approach permits a comprehensive analysis of users’ health and fitness behaviors, rendering it an ideal research context for a study seeking to delve into the broader implications of fitness applications. Second, Fitbit is renowned for its wearable fitness trackers, including Fitbit devices, which harmoniously synchronize with the Fitbit app, supplying users with real-time data and insights. This integration provides researchers with the opportunity to scrutinize how the synergy between an application and wearable technology influences user behavior, motivation, and resulting outcomes. Third, the Fitbit app incorporates social and community features that facilitate user connectivity, encouraging engagement in challenges and interactions with friends. This social component presents an intriguing research focus, allowing for an examination of how social interactions within the app impact user motivation and commitment to workout routines. Fourth, as an established presence in the fitness industry, Fitbit consistently updates its features and introduces new functionalities. Researchers can investigate the effects of these updates on user engagement and behavior trends over time. Fifth, owing to Fitbit’s widespread popularity, it serves as a valuable foundation for comparative studies alongside other fitness applications. Researchers can evaluate the distinguishing features of Fitbit in comparison to its competitors and analyze how these distinctions affect user behavior and outcomes. Lastly, Fitbit’s involvement in the broader health and wellness trends aligns with societal shifts towards healthier lifestyles [9,10]. Researchers can explore how app usage patterns correspond to these evolving health and wellness trends.

The reviews were gathered from the Android operating system users’ app store, and to achieve the research objective, only those available between October 2019 and April 2020 were included to identify and compare factors associated with fitness app satisfaction and dissatisfaction. During this period, the Fitbit app received a total of six minor updates, specifically focusing on software enhancements and troubleshooting. These updates, primarily addressing bug fixes and performance improvements, were released approximately every four weeks. Although the impact on Fitbit users and products may not have been significant, they contributed to maintaining the app’s functionality and user experience. To ensure research validity and minimize differences in versions of the app, the Android operating system was used. Only five-star (satisfied) and one-star (dissatisfied) reviews were included in the data set, which was refined and screened for incomplete and non-readable feedback using RapidMiner Studio. After screening, 67,467 qualifying reviews were included in the analysis, which were then divided based on reviewer satisfaction. The number of satisfied (five-star) reviewers was 38,803, while the number of dissatisfied (one-star) reviewers was 28,664. RapidMiner Studio helped us clean the online review data by removing irrelevant characters and stopwords, handling duplicates or missing values, converting text to lowercase, and applying techniques to normalize the reviews. To categorize, RapidMiner Studio used extract relevant features, converting reviews into numerical representations of term frequency-inverse document frequency (TF-IDF), which can capture the essence of the reviews and enable machine-learning algorithms to work effectively. 

### 3.3. Data Analysis Procedures and Corresponding Findings

To assess the importance of words in the corpus, TF-IDF was separately calculated for the satisfied and dissatisfied groups. According to Ramos [37], high TF-IDF values indicate a strong relationship of words in the document where they appear. Specifically, TF-IDF enables scholars to identify the significance of terms within each review, calculating a weight for each term based on its frequency. This analysis technique helps researchers emphasize terms that are both frequent within the review and relatively unique to it that highlight and categorize important concepts. In other words, this analysis approach can assist in conceptually and empirically understanding the topics, themes, or research areas covered in all reviews without having to carefully read the entire text of each review [37]. By analyzing content similarities, scholars can organize a large collection of online reviews into meaningful groups that help scholars to identify common themes. To remove less specific and over-fitted words, words that appeared in less than five percent and more than 95 percent of the sample were eliminated using the pruning method, which is effective for a condensed cluster structure [38]. The resulting keywords were 10 for the satisfied group and 36 for the dissatisfied group. Next, the sets of semantically similar words that occurred together within the same context (i.e., satisfied, dissatisfied) were clustered using the k-medoids algorithm, where each keyword is assigned to precisely one set of clusters [39]. Specifically, this study employed the maximum number of iterations, which is one type of the k-medoids algorithm’s parameters. This parameter enables scholars to determine the maximum number of iterations that the algorithm can conduct to converge to an integrated solution [39]. When the algorithm does not converge within the specified number of iterations, it terminates. This algorithm clusters by partitioning a dataset into k-clusters, where each cluster is represented by a medoid (i.e., a data point centrally located within the corresponding cluster).

After analyzing the data, the study found that 10 keywords from the satisfied group were divided into three attributional clusters. The first cluster was related to self-regulation, as users are interested in controlling their weight (keywords included Fit-bit-good-us-weight-year). These sub-attributes are aligned with the concept of self-regulation in the fitness app context that is based on individuals’ ability to monitor, control, and adjust goals and progress of their health condition and physical activity (e.g., weight and year) [40]. The second cluster was related to self-monitoring, as users track their daily exercise via the app (keywords included excellent-keep-time-track). These sub-keywords demonstrate tracking mechanisms to record (excellent) workout activities and to make decisions regarding health condition, which are essential for self-regulation in the fitness app context [40,41]. The third cluster was related to gratification, as users were satisfied with the app (keywords included great-love). This is because gratification in the fitness app context involves the fulfillment of needs and the subsequent positive feelings that come from that psychological fulfillment, such as love and pleasure [42]. These clusters are illustrated in Figure 1 and Table 1.

Similarly, the study found that 14 keywords from the dissatisfied group were divided into three attributional clusters. The sub-attributes are conceptually related to a failure to meet users’ expectations about paid services or technical features. The first cluster was related to paid services, as some premium services are only provided to paid users (keywords included Fit-bit-use-month-pay-year). The comments and words mean that paid services of the Fitbit app do not meet users’ expectations or provide users with substantial benefits, resulting in users’ dissatisfaction [3]. The second cluster was related to compatibility issues, as users often experience syncing and connecting problems with the app and wearable device (keywords included app-battery-trouble-unable). When encountering compatibility difficulties or issues, users are more likely to be dissatisfied with the fitness app and its corresponding devices [41]. The third cluster was related to functional issues, as users encountered problems with the app after an update or reset (keywords included app-can’t-log-reset-update-work). This is because users tend to expect technological advances whenever having an update or reset, leading to users’ dissatisfaction and frustration in the fitness app context [3]. These clusters are illustrated in Figure 2 and Table 2.

### 3.4. Follow-Up Assessments for Reliability and Data Interpretation

The authors conducted a comprehensive evaluation of the chosen keywords for each cluster. This evaluation involved rigorous scrutiny using two key metrics: silhouette scores, which indicated a moderate level of separation, and within-cluster sum of squares (WCSS), which pointed to an optimal number of clusters. The primary objective of these assessments was to gauge the quality of the clustering in this study, as some of the generated keywords did not align well with the conceptual framework of their respective clusters. To calculate the overall silhouette scores, the authors employed the following formula: S = the sum of the average distance from each data point to all other data points within the same cluster divided by the number of data points. Notably, all cluster scores fell within the range of 0.5 to 0.8, indicating an ideal scenario with well-separated clusters. Additionally, the authors evaluated the WCSS by summing the squared distances for all clusters. This was performed individually for each cluster, and the sums were aggregated across all clusters. It is worth noting that the WCSS value can be significantly affected by factors such as the number of clusters, data distribution, and clustering quality. Consequently, in this study, the focus was on identifying the minimum value of WCSS as a measure of successful clustering. The fact that all WCSS values in this study exceeded 0 demonstrated that each data point was appropriately assigned to a cluster containing only that specific data point.

In tandem with these statistical assessments, the authors sought external validation by enlisting the input of three professors specializing in management and marketing. These professors were asked to evaluate the conceptual appropriateness of the keywords assigned to each cluster, providing ratings ranging from 0 (indicating lack of conceptual appropriateness) to 3 points (signifying high conceptual appropriateness). Subsequently, the authors collected and averaged the ratings provided by the professors to determine whether they exceeded 2 points, which was considered the threshold for conceptual appropriateness. Keywords that received ratings exceeding 2 points were then selected for clustering and naming, thereby enhancing the overall reliability and credibility of the study’s findings.

In practice, this study gathered 5 five-star (i.e., representing satisfaction) and one-star (i.e., representing dissatisfaction) reviews. By analyzing common themes and elements present in 5-star reviews, the authors identified satisfiers as psychological aspects of users, such as self-regulation, self-monitoring, and gratification, and dissatisfiers as functional aspects of Fitbit, such as paid services, compatibility issues, and functional issues. In other words, users’ psychological aspects and the perceived functional aspects of their Fitbit device play a different role in forming either satisfaction or dissatisfaction. For instance, within our dataset of Fitbit reviews, we observed instances where users expressed satisfaction with their Fitbit devices despite the absence of certain functional features. These users often emphasized that their primary motivation for using the Fitbit was to meet basic tracking and psychological needs, such as monitoring their daily step count, tracking their sleep patterns, or staying motivated to maintain an active lifestyle. In these cases, the Fitbit device served as a valuable tool for personal goal setting and self-improvement. Users were willing to overlook the absence of specific functional attributes, such as advanced workout modes or compatibility with third-party apps, because the core functionalities aligned with their fundamental objectives.

On the other hand, it was noteworthy that some users raised complaints about functional issues with their Fitbit devices, even when the manufacturer had made functional improvements over time. These users often placed a strong emphasis on the device’s functional capabilities, such as accurate heart rate monitoring, GPS accuracy, or seamless synchronization with their smartphones. In such instances, even if Fitbit had addressed certain functional deficiencies through software updates or newer models, users continued to express dissatisfaction if their core functional expectations were not met. This highlights the significance of functional performance and reliability in shaping user perceptions, as users closely associate the device’s core functions with their overall experience and satisfaction. Overall, our study reveals the intricate interplay between psychological and functional factors in determining satisfaction and dissatisfaction among Fitbit users. It underscores the importance of recognizing the diverse priorities and expectations of users within the wearable technology landscape.

## 4. Discussion and Conclusions

The aim of this research was to determine the factors that influence users’ satisfaction and dissatisfaction with physical activity apps. To accomplish this, the study examined reviews of the Fitbit app on the Google Play Store and identified three factors that led to satisfaction and three factors that led to dissatisfaction with the app’s fitness features. The analysis revealed three sets of satisfiers—self-regulation, self-monitoring, and gratification—as well as three sets of dissatisfiers—the paid service attribute, compatibility attribute, and functional attribute. According to the findings, users are more satisfied with an app that provides self-regulation, self-monitoring, and gratification, while the absence of these factors does not necessarily lead to dissatisfaction. In contrast, users are very dissatisfied with apps that require additional payments and fail to meet their compatibility and functionality expectations, but these dissatisfiers alone do not improve user satisfaction. The study contributes to the fitness app literature by highlighting the specific factors that lead to user satisfaction and dissatisfaction. Although previous research in this area has focused on motivation [15,42], the empirical findings of this study offer insight into the factors that influence users’ satisfaction and dissatisfaction, potentially leading to increased long-term exercise via fitness apps [43]. 

Apart from the theoretical contribution, this study also offers practical insights for fitness app developers during the COVID-19 situation and beyond. Firstly, self-regulation can act as a preventive measure for avoiding health problems [44]. Self-regulation refers to the individual’s self-generated thoughts, emotions, and actions that are tailored to attain personal goals [45] (p. 14). Therefore, a fitness app should provide users with a comprehensive self-regulation framework that reflects their perceptions, attitudes, and behaviors concerning physical activity goals, body shape, and condition on a daily, weekly, and monthly basis. Secondly, fitness apps should include synchronized monitoring of physical data to promote a healthy lifestyle [8]. This is because self-monitoring by users creates positive perceptions and attitudes towards the app’s utility and helps them to continuously monitor their health performance, including physiological data, resulting in higher satisfaction with the fitness app [14,46]. Thirdly, the uses and gratification theory explains how gratifications meet the subjective intrinsic needs of sport participants [47]. Thus, the gratification that fitness app users derive from the virtual platform should contribute to their satisfaction with the overall experience. Additionally, the assessment and emotional aspects of uses and gratification, such as pleasure and enjoyment, can be significant determinants of user satisfaction and the feeling of belonging in an online platform via virtual competition and rewards [48].

Looking at the themes that emerged within the dissatisfier group and from a practical perspective, a fitness app requires extra money to be paid in order to provide users with premium services with more detailed guidance and insights from a variety of programs [8]. However, an incongruence between users’ expectation and perceived outcome from the extended premium services results in their dissatisfaction [22]. Therefore, a fitness app offers a certain free trial period for premium services to give users time to evaluate the services before making a payment, which leads to reduced dissatisfaction with the app. Second, many comments were related to an app’s incompatibility with wearable devices. Since more than 90% of wearable device owners access the mobile apps [49], compatibility issues may increase users’ dissatisfaction level as well as create a resistance to exercise. Also, an app’s functional attributes were found to be a major dissatisfier. According to the two-factor model, hygiene factors may include usability or functionality; for instance, basic features in a user interface work without technical errors [50]. Thus, fitness app developers should keep updating to resolve any technical issues and errors once they are reported from users. In order to do so, the fitness app developers should stay current on users’ comments and ratings about their app in virtual stores.

To summarize, this study provides valuable insights and empirical evidence that can guide future research in the fitness app domain. The study goes beyond identifying motivational factors and highlights other important factors that lead to user satisfaction, including self-regulation, self-monitoring, and gratification attributes. Furthermore, in addition to identifying satisfiers, this study also explores the app components that can lead to dissatisfaction, including paid services, compatibility, and functionality issues. By utilizing the themes of satisfiers and dissatisfiers, scholars in the physical activity field and fitness app industry practitioners can evaluate and develop more effective fitness apps that can enhance user satisfaction and thus maintain customer loyalty.

## 5. Limitations and Future Research Directions

The first limitation of this study is that it relied solely on user reviews from the Google Play Store. While this dataset provided valuable insights, it may not fully represent the entire user population, as it excludes iOS users and those who did not leave reviews. Future research could consider incorporating data from multiple sources, such as iOS app store reviews or data from Fitbit users who do not use app stores. Second, user reviews are inherently subjective and may be influenced by personal preferences, emotions, or specific experiences. This study acknowledges that the sentiment of the reviews may not capture the complete picture of app performance and user satisfaction. Future studies could combine sentiment analysis with objective measures to provide a more comprehensive assessment. Third, this research focused on reviews between October 2019 and April 2020, during which the Fitbit app received minor updates. These updates may not have captured significant changes in the app’s features or user experience. Longitudinal studies covering a wider time frame could provide insights into the app’s evolving user satisfaction. Also, there are several challenges linked to relying solely on user reviews. First, relying on user-generated content means we have limited control over the data quality, completeness, and accuracy. Future research might explore methods to validate user-generated content and incorporate additional data sources for verification. Second, user reviews raise privacy and ethical concerns, as they contain personal opinions and experiences. Ensuring the ethical use of such data and obtaining informed consent from users for research purposes is an ongoing challenge in this field.

This research provides future research directions in this field. First, future research needs to investigate long-term trends in user satisfaction with fitness apps like Fitbit to understand how satisfaction changes over time, considering factors like app updates, user engagement, and evolving user needs. In addition, future studies should extend the analysis to include user reviews from iOS and other platforms to provide a more comprehensive view of user sentiment and to identify potential platform-specific issues. Furthermore, future research may combine user reviews with quantitative usage data and surveys to triangulate findings and gain a deeper understanding of user experiences, satisfaction, and behavior.

In conclusion, our study provides valuable insights into user satisfaction with the Fitbit app based on user reviews. However, it is important to acknowledge the limitations associated with this approach and consider potential future research directions to enhance our understanding of the evolving landscape of fitness apps and user preferences. By addressing these limitations and exploring new avenues of research, we can contribute to the continuous improvement of fitness apps and the well-being of users.

## Figures and Tables

**Figure 1 behavsci-13-00782-f001:**
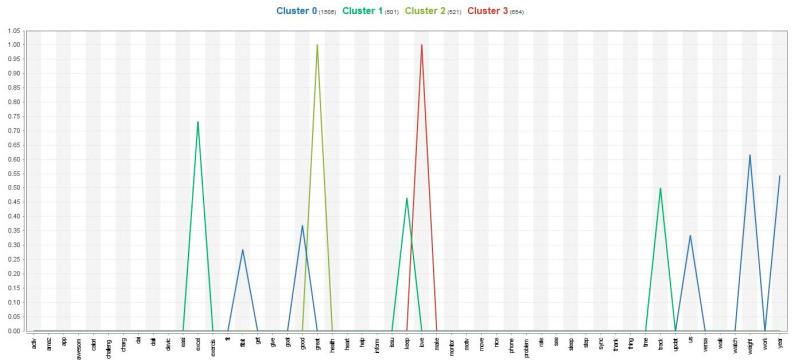
The centroid chart of the satisfied group.

**Figure 2 behavsci-13-00782-f002:**
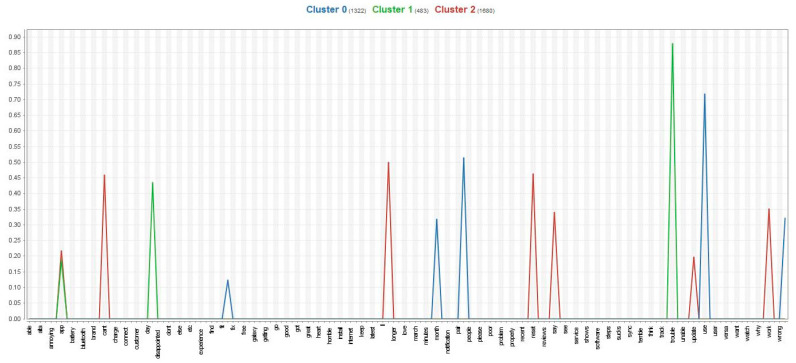
The centroid chart of the dissatisfied group.

**Table 1 behavsci-13-00782-t001:** The classified keywords for each cluster from the satisfied group.

Generated Keywords	First Cluster (Self-Regulation)	Second Cluster (Self-Monitoring)	Third Cluster (Gratification)
active, amaze, app, awesome, calori, challeng, dai, dali, devic, easi, excel, exercise, fit, fitbit, get, give, goal, good, great, health, heart, help, inform, issu, keep, love, make, monitor, motiv, move, nice, phone, problem, rate, see, sleep, step, sync, thank, thing, time, track, update, us, versa, walk, watch, weight, work, year	fit, fitbit, get, goal, good, great, update, us, versa, watch, weight, work, year	easi, excel, exercise, issu, keep, love, time, track, updat	good, great, health, keep, love, make

**Table 2 behavsci-13-00782-t002:** The classified keywords for each cluster from the dissatisfied group.

Generated Keywords	First Cluster (Paid Services)	Second Cluster (Compatibility Issues)	Third Cluster (Functional Issues)
able, ala, annoying, app, battery, Bluetooth, brand, cant, charge, connect, customer, day, disappointed, dont, else, etc., experience, find, fit, fix, free, galaxy, getting, go, good, got, great, heart, horrible, install, internet, keep, latest, longer, love, march, minutes, month, notification, pay, people, please, poor, problem, properly, recent, reset, reviews, say, see, service, shows, software, steps, sucks, sync, terrible, think, track, trouble, unable, update, use, user, versa, want, watch, why, work, wrong, year	fit, fix, minutes, month, notification, pay, people, update, use, user, work, wrong, year	annoying, app, battery, day, disappointed, track, trouble, unable	brand, cant, charge, longer, recent, reset, reviews, say, see, unable, update, use, why, work

## Data Availability

The data presented in this study are available on request from the corresponding author. The data are not publicly available due to privacy reasons.

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
