# Peer review of "Unpacking the Drivers of Dissatisfaction and Satisfaction in a Fitness Mobile Application"

_behavsci, 2023, doi:10.3390/bs13090782_

Round 1
Reviewer 1 Report
Dear authors,
Your dedication to exploring user satisfaction and dissatisfaction in fitness mobile applications is noteworthy, a valuable area of investigation particularly given the growing significance of technology in the fitness domain. The paper presents an interesting approach that utilizes Herzberg's two-factor model for this analysis. However, several critical aspects need to be addressed to strengthen the theoretical foundation and methodology of the study.
Consider condensing the abstract and introduction to present a concise overview of your research's objectives and significance. This would provide readers with a clear overview without overwhelming them with excessive details upfront. Instead, allocate space for a more expanded conceptual background section, explaining in depth why Herzberg's model suits the investigation of user satisfaction and dissatisfaction within the context of fitness mobile apps. In fact, while this model has been widely employed to measure job satisfaction, the rationale behind its suitability for this study remains unclear. To enhance the theoretical background, a more comprehensive discussion on why Herzberg's model was chosen and how its constructs align with the focus of your research is recommended. It would be beneficial to illustrate how this model has been successfully applied in previous computer science and mobile research contexts. Specifically, elucidate the aspects that render fitness mobile applications an apt subject for this analysis within the Herzberg framework.
Moreover, to expand the conceptual background it would be beneficial to delve deeper into the significance of online reviews, particularly how they shape consumer behavior and product improvement strategies.
The methodology and results sections are commendably detailed. However, certain portions could be further elucidated to ensure a comprehensive understanding for readers. Specifically, provide more detailed insights into the data collection process, including a step-by-step explanation of how the dataset of 100,000 English reviews from the Fitbit app was obtained using Python programming. Additionally, offer more robust reasoning for selecting the Fitbit app as your study's focal point, beyond just its popularity. Highlight its technological features that make it representative of fitness applications and elaborate on why the Google Play Store was chosen as the primary data source.
Please, address the discrepancy between the software-generated clusters and the attributional clusters mentioned among results. Providing a clear and detailed account of how the transformation from one type of clustering to another was executed will bolster the reliability of your findings. Please, note that the charts are too blurry and the keywords are not readable. Consider creating tables listing the keywords for each cluster and a system that makes it immediately clear how they were organized into the satisfaction and dissatisfaction factors. Moreover, expand upon your assertion that the absence of satisfaction factors doesn't necessarily lead to dissatisfaction. Offer a detailed account of the process that led to this conclusion, supplemented by concrete examples or data points from your analysis. Such clarification will enhance the credibility of your interpretations.
Create a distinct section to discuss the limitations of your study, including challenges linked to relying solely on user reviews. Additionally, delve into potential future research directions, thus concluding your study on a comprehensive note.
Reviewer 2 Report
Dear authors,
Thank you for the opportunity to review your research titled " Unpacking the Drivers of Dissatisfaction and Satisfaction in a 2 Fitness Mobile Application." It is an interesting study that suits the Behavioural Sciences topic. The research is original and interesting, but I consider major revisions as follows:
Theoretical framework
• The theoretical framework has focused on research on aspects of application use such as usability, usefulness, or gamification. However, when analyzing a mobile application, especially those related to health, I understand that more issues should be addressed besides those inherent to using said application. These questions may refer to helpful content from the point of view of the person's health and well-being. Are the functionalities and content of the application really of quality, that is, appropriate to provide both physical and emotional well-being? Has this issue been taken into account in the analysis or literature review? For example, consider nutrition, sports, Etc. Professionals design the content, and what has been the impact on the person's health and well-being after its use? Perhaps in the study you mentioned from Fur, Wu et al. [29], these critical aspects are considered and can complement or enrich your study.
• On the other hand, which factors of Herzberg's model correspond to possible factors of satisfaction or dissatisfaction with the application in these two senses: on the one hand, in the use of the mobile application under study, and on the other hand, in the content of the application from the point of view of improving health and well-being
Methodology
• The words on the graphics do not read well; they are too small
• I have a big concern about the content analysis. The methodology does not detail how the content analysis was carried out. One of the phases of this technique is to identify the variables and their corresponding categories with which each of the reviews that make up the unit of analysis will be coded (Nicholas et al., 2017). However, the research did not conduct this coding or subsequent analysis. The research aims to identify satisfaction and dissatisfaction factors through algorithms applied in text mining techniques such as K-Medoids clustering. Therefore, I understand that the methodology is not content analysis but the aforementioned "text mining" technique. Furthermore, it should be noted that other techniques to detect topics in texts, such as Latent Dirichlet Allocation, are used for topic modeling. K-medoid clustering is used for summary generation. According to Srivastava et al. (2022), an unsupervised extractive summarization approach with greater subtopic focus significantly improves over generic topic modeling semantics and deep learning approaches.
Nicholas, J., Fogarty, A. S., Boydell, K., & Christensen, H. (2017). The reviews are in: A qualitative content analysis of consumer perspectives on apps for bipolar disorder. Journal of Medical Internet Research, 19(4), e105-e105. https://doi.org/10.2196/jmir.7273
Srivastava, R., Singh, P., Rana, K.P.S., & Kumar, V. (2022). A topic modeled unsupervised approach to single document extractive text summarization. Knowledge-Based Systems, 246, 108636. https://doi.org/10.1016/j.knosys.2022.108636
Results
Since the article is based on the premise of applying Herzberg's theory theory, what relationship do the factors found have with it? The practical part should be more related to one of the article's main focuses, Herberg's theory.
Round 2
Reviewer 1 Report
I first want to express my sincere appreciation for the considerable effort that the authors have put into responding to previous concerns and suggestions. The extensive revisions made to the manuscript are indeed commendable, and it is evident that a considerable amount of time and dedication has been invested to improve the quality of the research.
I recognize that my initial recommendations may have required considerable effort, and I appreciate the authors' willingness to undertake the rigorous task of incorporating them into their work. Their commitment to refining the manuscript is truly remarkable.
At this second revision stage, I have only one additional suggestion: I recommend dividing sections 2 and 3 into subsections. The current length of these sections is considerable, and dividing them into subsections will improve the overall readability of the manuscript. Also, please ensure that the numbering of the sections is accurate, as I have noticed two instances of Section 3 in the current version.
Apart from this formal necessity, I believe that the improvements made to the manuscript have significantly improved its overall quality.
